# Experimental Investigation on the Mechanical Properties of Self-Compacting Concrete under Uniaxial and Triaxial Stress

**DOI:** 10.3390/ma13081830

**Published:** 2020-04-13

**Authors:** Hongbo Li, Jianguang Yin, Pengfei Yan, Hao Sun, Qingqing Wan

**Affiliations:** 1College of Civil and Hydraulic Engineering, Ningxia University, Yinchuan 750021, China; yinjianguang@nxu.edu.cn (J.Y.); 12019130763@nxu.edu.cn (P.Y.); 12018130640@nxu.edu.cn (H.S.); 20181306401@nxu.edu.cn (Q.W.); 2Ningxia Research Center of Technology on Water-saving Irrigation and Water Resources Regulation, Yinchuan 750021, China; 3Engineering Research Center for Efficient Utilization of water Resources in Modern Agriculture in Arid Regions, Yinchuan 750021, China

**Keywords:** self-compacting concrete, fly ash, mechanical properties, triaxial test, failure criterion

## Abstract

To explore the influence of fly ash (FA) and silica fume (SF) on the mechanical properties of self-compacting concrete (SCC) under uniaxial and triaxial, the compressive strength test, splitting strength test, ultrasonic testing test, and triaxial test were performed in this paper. The results show that the 3 days compressive strength and splitting strength of SCC decreased with the increase of FA substitution rate. The 28 days, 56 days, and 91 days compressive strength and splitting strength of SCC increased first and then decreased with the increase of FA substitution rate. The peak stress and peak strain of SCC gradually increased with the increase of confining pressure. The peak stress and strain of SCC increased first and then decreased with the increase of FA substitution rate. Moreover, the relationship models between compressive strength and splitting strength, between compressive strength and amplitude, between peak stress, peak strain and confining pressure under different FA substitution rates were proposed. As a conclusion, the addition of SF can increase the strength of SCC obviously. Under uniaxial stress, SCC failure mode is splitting failure, under triaxial stress, SCC failure mode is shear failure. Based on the Mohr-Coulomb strength theory, the failure criterion of SCC with FA and SF was discussed.

## 1. Introduction

Self-compacting concrete (SCC), also known as self-leveling concrete, is a kind of concrete which can fill all parts of concrete formwork and steel bar gap compactly and uniformly without vibration or slightly vibration under the action of its own gravity [1]. Compared with ordinary concrete, it has the characteristics of high cohesion, fluidity, anti-segregation and vibration-free. The deviation of steel bar, formwork and reserved embedded parts caused by vibration can be avoided in the process of construction and pouring. At the same time, it can reduce noise pollution, reduce labor cost, and improve engineering efficiency. In recent years, with the development of construction machinery and technology, it was widely used in construction, bridge, tunnel, hydraulic engineering, and other fields [2,3,4,5,6,7]. To obtain better performance, various mineral admixtures were usually added to SCC to obtain better performance. Esfandiari et al. [8] studied the effect of perlite powder and silica fume (SF) on SCC with lime-cement binder and found that the addition of perlite powder and SF can accelerate the formation of C-S-H gel in SCC and improve compressive strength. Faraj et al. [9] found that the ductility, fracture energy and durability of self-compacting high strength concrete can be improved by adding recycled polypropylene particles. Ling, Akcay et al. [10,11] studied the effect of metakaolin (MK) and SF on SCC, found that the addition of MK and SF to SCC can improve the pore structure, strength and durability, but it has a negative influence on fracture energy. Gupta et al. [12] found that the addition of copper slag can improve the cohesion of SCC cement particles, make the structure of SCC more compact and improve the strength.

At present, the output of fly ash (FA) in China is about 550 million tons [13], and the annual output of SF is about 400,000 tons. With the development of electric power and metallurgical industry, the output of FA and SF is still increasing year by year. A large amount of FA and SF accumulate together, which not only pollutes the environment, but also harms human health. However, with the deepening of research, they have been used as mineral admixtures commonly in the preparation of concrete (including cement mortar). The use of FA and SF cannot only reduce the emission of greenhouse gases and solid waste in the production of cement clinker, improve the use of resources, but also improve the comprehensive properties of concrete and cement mortar [14,15,16]. Yang, Shi et al. [17,18] studied the effect of FA on the mechanical properties of concrete, found that the concrete prepared with FA partially replacing cement had lower early strength but higher later strength. Li et al. [19] studied the effect of SF on the compressive strength of rubber concrete, found that the early strength of concrete prepared with SF instead of cement can be improved, but the strength increases slowly in the later stage. Therefore, the use of double mixing technology provides a green channel for the use of FA and SF, so that the respective characteristics of FA and SF can be brought into full play.

The good working performance of SCC is the basic premise to ensure the mechanical properties and durability of concrete, and the safe and reliable mechanical performance is an indispensable condition to ensure concrete. Therefore, under the premise of good working performance, reliable mechanical properties can ensure the high durability of concrete. Thus, it can be seen that the working performance is the premise, the durability is the result, and the mechanical properties are guaranteed to meet the durability requirements under the working performance. Therefore, mechanical properties have always been the focus of concrete research. At present, domestic and foreign scholars have conducted a large number of experimental studies on the mechanical properties of SCC. Mohammed et al. [20] studied the effect of recycled concrete aggregate on the mechanical strength, flexural properties, and fracture energy of SCC, and found that although the addition of recycled aggregate concrete aggregate reduces the mechanical properties of concrete and the mechanical properties of test beams, the strength can still meet the results. Carballosa et al. [21] found that adding 10% K-type calcium sulphoaluminate expansive agent to SCC can improve the early compressive and flexural strength of SCC, but when the content of K-type calcium sulphoaluminate expansive agent increased to 15%, it has a negative effect on the compressive strength and apparent strength of SCC. Zhang et al. [22] carried out biaxial experimental study on SCC with FA, the results show that compared with ordinary concrete under biaxial loading, the principal stress of SCC was affected by lateral compressive stress, and the strength criterion equation of SCC under biaxial loading was put forward. Sun et al. [23] carried out an experimental study on the composite mechanical properties of self-compacting lightweight aggregate concrete, the results show that under the combined action of compression and shear, the shear failure load, residual load and shear failure displacement of SCC increase gradually with the increase of axial compression ratio, and the unified failure criterion equation of concrete compression-shear composite action was put forward. Wang et al. [24] conducted an experimental study on the compression-shear performance of SCC, found that the shear strength of SCC under combined compression-shear stress can be divided into common bond stress and residual friction stress, and the unified strength failure criterion and the failure criterion based on octahedral stress space were proposed. From the present research situation, we can see that scholars have made a series of achievements in the mechanical properties of SCC. However, there are few research results about the time-varying change law of mechanical properties under uniaxial stress and the change law of mechanical properties under triaxial stress of SCC with FA and SF, especially the strength calculation model of SCC with FA and SF under triaxial stress. In practical engineering, concrete was subjected to multiaxial stress in many cases, such as dams, marine engineering, and concrete at beam-column joints. Therefore, it is necessary to study the change law of mechanical properties of SCC with FA and SF under triaxial stress and the time-varying change law of mechanical properties under uniaxial stress.

This paper mainly discussed the effects of 4% SF and different substitution of FA on the mechanical properties of SCC under uniaxial stress and triaxial stress, which provided reference for the theoretical research and engineering application of SCC with FA and SF. The time-varying change law of mechanical properties of SCC under uniaxial stress was studied by compressive strength test and splitting strength test. The triaxial test was carried out to analyze the change law of mechanical properties of SCC under different confining pressure and to discuss the failure criterion of SCC strength. The relationship between amplitude and compressive strength was analyzed by ultrasonic test.

## 2. Materials and Methods

### 2.1. Materials

In this study, cement used was Ningxia horse racing P·O 42.5R (Ningxia Horse Racing Cement Co., Ltd, Yinchuan, China). The grade П FA was produced by Yinchuan Thermal Power Plant (Yinchuan, China). The main performance indexes of cement and FA are shown in Table 1. Local SF in Yinchuan was adopted, and its main performance indexes are shown Table 2. The fine aggregate was medium sand with a bulk density of 1762 kg/m^3^ and a fineness modulus of 2.63. The coarse aggregate used was continuously graded gravel in Yinchuan. UEA concrete expansion agent produced in Yantai, Shandong Province, China was adopted. Polycarboxylic acid superplasticizer was used with a water reducing rate of 30%. Tap water was used.

### 2.2. Mix Proportion

According to the provisions and requirements of SCC in The Technical Specification For Application of Self-compacting Concrete (JGJ/T283-2012) [25], and considered the influence of FA and SF, a total of 10 groups of SCC with design strength of C50 were prepared. Among them, the mix proportion of FSCC0 and SFSCC0 were used as SCC control group. The total substitution of SCC cementitious materials remained unchanged, the water binder ratio was 0.272, the sand rate was 47.68%, and the water reducing agent substitution was 1.19% of cementitious material. The quality of cement substituted by FA was 0%, 10%, 20%, 30%, and 40% respectively, the quality of cement substituted by SF was 4%, the test mix proportion is shown in Table 3.

SCC was prepared according to the test mix proportion, two types of test specimens with dimensions of 100 mm × 100 mm × 100 mm and Φ 50 mm × 100 mm were made after poured and formed. After the specimen was formed for 24 h, the mold was removed and put into the HBY-40B cement constant temperature and humidity standard curing box (Shangyu Weiying Geotechnical Instrument Factory, Shaoxing, China) (relative humidity ≥ 95%, temperature 20 ± 2 °C).

### 2.3. Methods

Compressive strength and splitting strength test. The compressive strength and splitting strength of SCC were tested according to The Standard for Test Methods for Mechanical Properties of Ordinary Concrete (GB/T50081-2002) [26]. The test included five ages, which were 3 days, 7 days, 28 days, 56 days, and 91 days, respectively. The dimensions of the specimen was 100 mm × 100 mm × 100 mm. Compressive strength test was carried out on the SHT4106 microcomputer controlled electro-hydraulic servo universal testing machine (MTS Systems (China) Co., Ltd, Shanghai, China) and the loading rate was 0.7 MPa/s during the test. The splitting strength test was carried out on the CMT5305 microcomputer controlled electronic universal testing machine (MTS Systems (China) Co., Ltd, Shanghai, China) and the loading rate was 0.07 MPa/s during the test.

Ultrasonic testing test. Ultrasonic testing of specimens was determined by NM-4A non-metallic ultrasonic testing analyzer (Figure 1) produced by Beijing Koncrete Testing Technology Co., Ltd., (Beijing, China). The instrument should be zeroed before testing, the purpose of zeroing was to eliminate the sound delay of the instrument transmitter and receiver transducer system in the sonic time test. During the measurement, five measuring points as shown in Figure 2 were arranged on four sides of the specimen. Then the transmitter and transducer probe were coated with butter coupling agent, the probe and the concrete surface of the measuring point were in a good coupling state. After pressed the sampling key, the instrument began to emit ultrasonic wave and sample.

Triaxial test. The test was carried out on the YY-RBSZ-1000 rock triaxial (creep) tester developed by Rugao City Yuanye Exploration machinery Co. Ltd, (Rugao, China). Figure 3a shows the test device. At the top of the instrument, there were small upper and lower oil cylinders in the confining pressure chamber, which were used to lift or lower the confining pressure chamber. The confining pressure chamber was connected to the oil pressure pump station through a special oil pipe, which was responsible for placing test blocks and providing confining pressure. The upper end of the confining pressure chamber was equipped with a sensor column and a 1000 KN pressure sensor, which can directly measure the pressure on the specimen. A displacement meter was installed on the column and the middle crossbeam to measure the deformation of the specimen. Four different confining pressure values of 0 MPa, 5 MPa, 10 MPa, and 15 MPa were selected for the test. The age of the specimen was 28 days and the size was Φ 50 mm × 100 mm. During the test, the lateral confining pressure was applied at the rate of 0.05 MPa/s according to the predetermined design value, then kept the specimen in the state of hydrostatic pressure (*σ*_1_ = *σ*_2_ = *σ*_3_). After reached the predetermined design value, kept the lateral confining pressure constant. The axial load adopts the load control loading system, and loaded at the rate of 0.5 MPa/s until the specimen was damaged. Figure 4 shows the stress model and loading path of the specimen.

## 3. Results and Discussions

### 3.1. Compressive Strength

As the basic index value of mechanical properties of SCC, compressive strength was also the basis for the study of other mechanical indexes. The results of SCC compressive strength test at different ages are shown in Figure 5. The average values and standard deviation of the test are shown in Table 4 and Table 5.

As shown in Figure 5, the addition of FA and SF has obvious effect on the development of compressive strength of SCC at different ages. The 3 days compressive strength of SCC decreased gradually with the increase of FA substitution rate and were lower than the SCC compressive strength of the control group (SFSCC0) of the same age. This is because the activity of FA was lower than that of cement, the reaction in the early stage of hydration was slowed, and its activity cannot be brought into full play in a short time [27], the early strength of SCC was mainly borne by cement. With the increase of FA substitution rate, the substitution rate of cement decreased, which led to the decreased of SCC strength. The 28 days compressive strength of SFSCC20 reached the peak value of 72.95 MPa of this age, which was 10.7% higher than that of the control group (SFSCC0) of the same age. The 56 days and 91 days compressive strength of SFSCC20 reached the peak value of 80.13 MPa and 87.59 MPa, which increased by 7.38% and 8.9% respectively compared with FSCC20 of the same age. The 28 days, 56 days, and 91 days compressive strength of SCC increased first and then decreases with the increase of FA substitution rate. The main reason was that the activity of FA gradually developed with the increase of age, and the SiO_2_ and Al_2_O_3_ in FA reacted with Ca(OH)_2_ produced by cement hydration to form dense C-S-H and C-A-H gels [28], namely:(1)$$\begin{array}{l} \;\mathrm{Si}O_{2}+x\mathrm{Ca}(\mathrm{OH}{)}_{2}+(y-1)H_{2}O\to x\mathrm{CaO}•\mathrm{Si}O_{2}•yH_{2}O\\ Al_{2}O_{3}+x\mathrm{Ca}(\mathrm{OH}{)}_{2}+(y-1)H_{2}O\to x\mathrm{CaO}•Al_{2}O_{3}•yH_{2}O \end{array}$$

C-S-H and C-A-H gels tighten the relationship between FA particles and cement particles and improved the compactness of the internal structure of SCC. Meanwhile, FA particles and SF particles are smaller than cement particles and can be filled between cement particles to further improved the stability and compactness of SCC. Therefore, the compressive strength of SCC increased with the increase of FA substitution rate. With the further increase of FA substitution rate, the degree of secondary reaction of hydration increased and the consumption of Ca(OH)_2_ increased correspondingly, while the decrease of the proportion of cement led to the decrease of the amount of Ca(OH)_2_ produced by hydration, which slowed down the overall rate of secondary hydration. As a result, the connection between C-S-H and C-A-H gels particles was not dense enough, which reduced the compressive strength of SCC.

When the substitution rate of FA was the same, the compressive strength of SCC with FA and SF was higher than that of SCC with FA at the same age. On the one hand, because the particle size of SF was smaller than that of FA and cement particles, it can form a micro-filling effect between FA and cement particles and improve the compactness of SCC. On the other hand, because silica fume has strong pozzolanic activity, it can quickly react with Ca(OH)_2_ produced by cement hydration to form C-S-H gel. C-S-H gel can improve the pore structure and make the microstructure of SCC more compact [29], thus improved the compressive strength. In summary, SFSCC20 achieved the best FA substitution rate, its strength and stability were better than other SCC with FA and SCC with FA and SF, the compressive strength of 72.95 MPa, 79.13 MPa and 87.59 MPa were obtained at 28 days, 56 days and 91 days respectively. While ensuring the requirement of compressive strength, in order to increase the use FA resources as much as possible, it is suggested that the replacement rate of FA should be 20%.

### 3.2. Splitting Strength

The splitting strength test results of SCC at different ages are shown in Figure 6. The average values and standard deviation of the test are shown in Table 6 and Table 7.

According to the result, the effect of FA substitution rate on the splitting strength of SCC varies with ages. The 3 days splitting strength of SCC decreased with the increase of FA substitution rate, compared with control SCC of the same age, the strength decreased by 3.56–33.62%, the change law and the cause were the same as those of the compressive strength test.

The 7 days splitting strength of SCC with FA increased by 5.07–10.63% compared with 3 days, and that of SCC with FA and SF increased by 8.25–23.63% compared with 3 days. Thus, it can be seen that the addition of SF has a significant effect on the increase of early strength of SCC. This is because SF particles were very fine and had strong pozzolanic activity, which can control the reaction of alkali aggregate. Its main component SiO_2_ can react to consume Ca(OH)_2_ produced by hydration of cement to produce C-S-H gel material, and improved the splitting strength of SCC. The 56 days splitting strength of SFSCC20 reached the peak value of 7.25 MPa of this age, which was 9.02% higher than that of 28 days SFSCC20. The 91 days splitting strength of SFSCC20 reached the peak value of 7.97 MPa of this age, compared with the splitting strength of SFSCC20 at 28 days and 56 days, splitting strength increased by 18.07% and 9.93% respectively.

The 28 days, 56 days and 91 days splitting strength of SCC increased first and then decreased with the increase of FA substitution rate. On the one hand, this trend was caused by the failure mode of splitting test specimen. The splitting failure extended from the microcracks in the aggregate transition zone to both sides, and deflected when it met the interface transition zone, resulted in the formation of a main crack and damaged. When FA substitution rate increased, the interface transition zone increased, and the continuous migration of microcracks consumed more fracture energy, thus improved the splitting strength of SCC. On the other hand, due to the increase of FA content, the filling effect of FA and the efficiency of secondary hydration reaction were improved, the compactness of SCC increased, and the splitting strength of SCC was improved. As FA substitution rate continues to increase, the proportion of cement clinker was correspondingly reduced, and there was not enough Ca(OH)_2_ to excited FA in SCC, the increase of the amount of unreacted FA deteriorated the pore structure of SCC, thus reduced the splitting strength of SCC [30,31].

### 3.3. Relationship Between Compressive Strength and Splitting Strength

Splitting strength as another important index value of mechanical properties of concrete, the relationship between it and compressive strength was also the focus of people’s research. The conversion relation between compressive strength and splitting strength of ordinary concrete commonly used in China [32] is shown in Equation (2).
(2)$$f_{t}=0.19(f_{\mathrm{cu}}{}^{\frac{3}{4}})$$
where *f*_t_ is the splitting strength of ordinary concrete, MPa; *f*_cu_ is the compressive strength of ordinary concrete, MPa.

The relationship between the average splitting tensile strength of ordinary concrete and the prescribed compressive strength of a cylinder given by ACI318-08 [33] in the United States is shown in Equation (3).
(3)$$f_{\mathrm{ctm}}=0.556\sqrt{f_{c}\text{′}}$$
where *f*_ctm_ is the average splitting tensile strength of ordinary concrete, MPa; *f*_c_’ is the compressive strength of cylinder, MPa.

Convert the cylinder compressive strength *f*_c_’ and the average cylinder compressive strength *f*_cm_’ according to Table 5.3.2.1 in ACI 318-08, and the cylinder compressive strength and cubic compressive strength are converted according to the European standard EN 1992-1. Approximately transformed into the power function relationship between the average splitting tensile strength and the cube compressive strength [34] shown in Equation (4).
(4)$$f_{\mathrm{ctm}}=0.3f_{c\mathrm{um}}{}^{0.6}$$
where *f*_cum_ is the average compressive strength of cube, MPa.

According to the test results, the development characteristics of compressive strength of SCC and ordinary concrete were different. Therefore, the strength conversion formulas of ordinary concrete were no longer applicable to SCC. Through the analysis of the test results, the relationship between splitting strength and compressive strength of SCC was obtained as shown in Figure 7. After fitted the test data with power function and linear function, Equations (5) and (6) were obtained.

The fitted relations are as follows:(5)$$\begin{array}{l} f_{t}=0.633+0.083f_{\mathrm{cu}}\\ R^{2}=0.992 \end{array}$$
(6)$$\begin{array}{l} \;f_{t}=0.150(f_{\mathrm{cu}}{)}^{0.886}\\ R^{2}=0.990 \end{array}$$
where *f*_t_ is the splitting strength of ordinary concrete, MPa; *f*_cu_ is the compressive strength of ordinary concrete, MPa.

Through the fitted results, it can be seen that the two relations were in good agreement with the experimental data. Equations (5) and (6) can accurately reflect the conversion relationship between SCC compressive strength and splitting strength. Based on the experience of used power function in ordinary concrete, it is suggested that Equations (5) and (6) be used to predict the splitting strength of SCC.

### 3.4. Ultrasonic Test

After sampling, the measured sound wave was converted into frequency spectrum by FTT, and the peak value of spectrum amplitude was taken as the test value. The spectrum of SCC is shown in Figure 8.

According to the test results, The relationship between amplitude and compressive strength of SCC with FA and SF was obtained as shown in Figure 9.

The following calculation formula can be obtained by fitted:(7)$$\begin{array}{l} y=-27.680+2.094x\\ R^{2}=0.999 \end{array}$$

As can be seen from the fitting results, the fitting curve was in good agreement with the measured value. Then Equation (7) can be well used to reflect the relation between compressive strength and amplitude of SCC.

### 3.5. Triaxial Test

#### 3.5.1. Failure Characteristics

The failure modes of SCC specimens under different confining pressures are shown in Figure 10.

By observing the crack direction and shape of the specimen, it can be found that the failure mode of the specimen has little to do with the substitution rate of FA and SF, but was mainly affected by the confining pressure. When the confining pressure value *σ*_2_ = *σ*_3_ = 0 MPa (under uniaxial stress), the failure mode of the specimen was splitting failure, and the specimen after failure basically maintains its original integrity, and there was no obvious spalling of concrete fragments. In addition, there were many cracks on the failure surface of the specimen, and some cracks penetrate both ends of the specimen, which made the specimen lost the ability to continue to bear load, thus caused damage to the specimen. When the confining pressure value *σ*_2_ = *σ*_3_ = 5 MPa, the failure mode of specimen changed from splitting failure to shear failure, and the mortar and coarse aggregate on the failure surface were cut and accompanied by the spalling of concrete. There was a main crack on the failure surface of the specimen from top to bottom, and the crack direction was no longer parallel to the direction of the principal stress, but at an angle of about 5° to 10° with the direction of the principal stress. When the confining pressure value *σ*_2_ = *σ*_3_ = 10 MPa, the crack angle varied in the range of 20°–30°, and there was a main crack on the failure surface that ran through both ends of the specimen. Moreover, the concrete at the penetration has staggered with each other due to shear action, and there was a slight bulge in the middle of the specimen. When the confining pressure value *σ*_2_ = *σ*_3_ = 15 MPa, there was a main crack on the failure surface of the specimen, which was about 45° to the direction of the principal stress. The concrete on both sides of the main crack was sheared with each other, and there was crushed concrete powder on the failure surface.

#### 3.5.2. Stress-Strain Curve

The load-displacement data were collected by YY-RBSZ-1000 rock triaxial (creep) tester, and converted into stress and strain values by Equation (8), and drew the *σ*-*ε* curve as shown in Figure 11.
(8)$$\sigma =\frac{P}{A_{a}}\;\varepsilon =\frac{\Delta l}{H}\;A_{a}=\frac{A_{0}}{1-\varepsilon }$$
where *σ* is the axial stress value of SCC under different confining pressure, MPa; *P* is the axial load, N; *A*_a_ is the cross sectional area of the specimen in the process of shearing, mm; *ε* is the axial strain; *Δl* is the axial compression deformation of specimen, mm; *H* is the height of specimen, mm; and *A*_0_ is the initial cross sectional area of specimen, mm^2^.

As shown in Figure 11, at the initial stage of loading, as the load increases, the slope of the stress-strain curve of the specimen was basically unchanged and approximately linear increase, showed elastic deformation. With the increase of the load, the specimen began to enter the stage of plastic deformation, in which the strain developed faster than the stress, and the slope of the stress-strain curve decreased gradually. The confining pressure has an obvious effect on the stress-strain curve of the specimen. When the confining pressure value *σ*_2_ = *σ*_3_ = 0 MPa, the stress-strain curve of the specimen has obvious peak, the peak stress and the peak strain are smaller than those of the specimen with confining pressure, and the curve of the descending section was steeper, which indicated that the plastic deformation of the concrete was small. With the increase of confining pressure, the peak value of the curve increased gradually, while the peak became smooth, and the peak stress and peak strain increased gradually. When the confining pressure value *σ*_2_ = *σ*_3_ = 5 MPa, the peak of the curve can be seen clearly, but the slope of the curve in the descending section decreased gradually. When the confining pressure value *σ*_2_ = *σ*_3_ = 10 MPa, the peak of the curve was no longer obvious, and the curve became smoother after the peak, showing good ductility. which showed a good ductility characteristic. When the confining pressure value *σ*_2_ = *σ*_3_ = 15 MPa, the spike no longer existed, and there was basically no yield drop in the curve after the peak. This phenomenon can be seen well from the comparison of the stress-strain curve in Figure 11.

#### 3.5.3. Peak Stress

According to the stress-strain curve of the specimen, the peak stress of SCC under different confining pressures can be obtained as shown in Figure 12.

As can be seen from Figure 12, the confining pressure has a significant effect on the peak stress. Under triaxial stress, the peak stress of SCC was above 80 MPa, which was much higher than that of uniaxial stress, and the peak stress of SCC increased gradually with the increase of confining pressure. This phenomenon was due to the fact that the action of confining pressure limited the expansion of microcracks in the specimen, weakened the negative effect of microcracks on the strength of the specimen under triaxial stress, and improved the strength of the specimen. Moreover, the greater the confining pressure, the more obvious the improvement effect. After dimensionless processing of test data, the dimensionless relationship between peak stress and confining pressure was obtained (Figure 13).

The following relation was obtained by fitting:(9)$$\frac{\sigma_{1}}{\sigma_{0}}=1+A\frac{\sigma_{2}}{\sigma_{0}}$$
where *σ*_1_ is the peak stress of SCC under different confining pressure, MPa; *σ*_0_ is the peak stress of SCC under uniaxial stress (*σ*_2_ = *σ*_3_ = 0 MPa), MPa; *σ*_2_ is the confining pressure, MPa; and A is the coefficient related to FA substitution rate *γ*.

Table 8 shows the relation after fitting of SCC. It can be seen that there was a certain relationship between coefficient A and FA substitution rate *γ*. The fitting curve shown in Figure 14 can be obtained after fitting and analysis the test data.

According to the fitting results in Figure 14, Equation (10) can be obtained.
(10)$$A=6.601+0.054\gamma -0.002\gamma^{2}$$

Bring Equation (10) into Equation (9), the relation of peak stress and confining pressure of SCC shown in Equation (11) can be obtained.
(11)$$\frac{\sigma_{1}}{\sigma_{0}}=1+(6.601+0.054\gamma -0.002\gamma^{2})\frac{\sigma_{2}}{\sigma_{0}}$$

#### 3.5.4. Peak Strain

Peak strain was an important part in the study of mechanical properties of SCC under triaxial stress, which was of great significance for in-depth analysis of deformation and failure characteristics of specimens. The strain of SCC peak under different confining pressure is shown in Figure 15.

As can be seen from Figure 15, the confining pressure has a great influence on the peak strain of SCC, the peak strain under triaxial stress was significantly higher than that under uniaxial stress. Moreover, the larger the confining pressure was, the more significant the effect on the peak strain was. This showed that the existence of confining pressure can restrain the development of cracks and improved the deformation ability of the specimen [35,36], and the larger the confining pressure was, the more obvious the inhibition was. After dimensionless processing of test data, the dimensionless relationship between peak strain and confining pressure values can be obtained (Figure 16), Equation (12) can be obtained by fitting.
(12)$$\frac{\varepsilon }{\varepsilon_{0}}=1+B\frac{\sigma_{2}}{\sigma_{0}}$$
where *ε* is the peak strain of SCC under different confining pressures; *ε*_0_ is the peak strain of SCC under uniaxial stress (*σ*_2_ = *σ*_3_ = 0 MPa); and B is the coefficient related to FA substitution rate *γ*.

The relationship between coefficient B and FA substitution rate γ is shown in Table 9, and the fitting curve shown in Figure 17 can be obtained by fitting the data in Table 9.

According to the fitting results in Figure 17, the following relation can be obtained.
(13)$$B=14.245+0.075\gamma -0.002\gamma^{2}$$

Bring Equation (13) into Equation (12), the stress-strain and confining pressure relation shown in Equation (14) can be obtained.
(14)$$\frac{\varepsilon }{\varepsilon_{0}}=1+(14.245+0.075\gamma -0.002\gamma^{2})\frac{\sigma_{2}}{\sigma_{0}}$$

#### 3.5.5. Failure Criteria

Many strength theories have been obtained in the study of mechanical properties of ordinary concrete, among which the strength theory proposed by Mohr-Coulomb was typical. Based on Mohr-Coulomb strength theory, this paper discussed the failure criterion of SCC with FA and SF. According to Mohr-Coulomb strength theory, the compressive failure state of the material was controlled not only by the shear stress *τ*, but also by the normal stress *σ* on the shear plane [37]. The relation is as follows:(15)τ=c+σtanφ
where *τ* is the shear stress, MPa; *c* is the cohesion, MPa; *σ* is the normal stress, MPa; and the *φ* is internal friction angle, °.

The Mohr-Coulomb strength theory expressed the failure criterion of concrete by stress circle envelope. When the concrete was in the limit equilibrium state, the stress circle was called the failure stress circle, and the envelope (shear strength line) was tangent to the stress circle, while the corresponding plane was called shear failure plane. When the concrete failed to reach the stress limit state, intercross between envelope and stress circle. The Mohr-Coulomb strength theory was used to process the data of Figure 12, and the Mohr-Coulomb parameters were calculated by drawing the stress circle (Figure 18). The cohesion *c* and internal friction angle *φ* of 28 d SCC with FA and SF under triaxial stress were obtained (Table 10). It can be seen that the cohesion and internal friction angle of SCC increased first and then decreased with the increase of FA substitution rate. This was because the filling effect and the degree of secondary hydration of FA were improved with the increase of FA replacement rate, which increased the cohesion between particles and the angle of internal friction. With the continuous increase of FA substitution rate, the secondary hydration reaction conditions became worse and the overall hydration rate slowed down, so that the connection between the particles of the hydration products was not closed enough, which reduced the cohesion and internal friction angle.

After dimensionless processing of test data, the dimensionless diagram shown in Figure 19 was obtained. The relation (14) of shear stress *τ* was obtained by fitting analysis:
(16)$$\frac{\tau }{\sigma_{0}}=D+E(\frac{\sigma }{\sigma_{0}})$$
where *σ*_0_ is the peak stress of SCC under uniaxial stress (*σ*_2_ = *σ*_3_ = 0 MPa), MPa; Coefficient D is the ratio of cohesion *c* to *σ*_0_; and E is the fitting coefficient.

The E = 1.091 was obtained by the least square method, then the empirical relation of shear strength of SCC with FA-SF can be obtained as follows:(17)$$\frac{\tau }{\sigma_{0}}=D+1.091(\frac{\sigma }{\sigma_{0}})$$

Through the regression analysis of the FA substitution rate *γ* and the coefficient D of the test data in Table 11, the fitting results shown in Figure 20 can be obtained.

The following relationships can be obtained.
(18)$$\begin{array}{l} D=0.2046-0.0002-0.3707\gamma^{2}+1.1890\gamma^{3}\\ R^{2}=0.999 \end{array}$$

Then the failure criterion relation of SCC with FA and SF is as follows:(19)$$\frac{\tau }{\sigma_{0}}=0.2046-0.0002\gamma -3.3707\gamma^{2}+1.1890\gamma^{3}+1.0905(\frac{\sigma }{\sigma_{0}})$$

It can be seen from Figure 19 that the fitting curve was in good agreement with the measured value, thus Equation (19) has a good applicability to the expression of the failure law of SCC with FA and SF.

## 4. Conclusions

In this paper, the compressive strength test, splitting strength test, ultrasonic testing test and triaxial test of SCC were carried out, respectively. The following conclusions are drawn:The 3 days compressive strength and splitting strength of SCC decreased with the increase of FA substitution rate. The 28 days, 56 days, and 91 days compressive strength and splitting strength of SCC increased first and then decreased with the increase of FA substitution rate. When the FA substitution rate is 20% and SF substitution rate is 4%, the strength of SCC reached the maximum.According to the test results, the relationship model between compressive strength and splitting strength was proposed, the relationship model between amplitude and compressive strength was proposed, and the relationship model of peak stress, peak strain and confining pressure value of SCC with FA and SF under different FA substitution rate was proposed.The failure modes of SCC were greatly affected by confining pressure. When confining pressure = 0 MP, the failure mode is splitting failure. When the confining pressure increased to 5–15 MPa, the failure mode changed to shear failure.Based on the Mohr-Coulomb strength theory, the failure criterion of SCC with FA and SF was discussed, the fitting results were well good agreement with the measured values.

## Figures and Tables

**Figure 1 materials-13-01830-f001:**
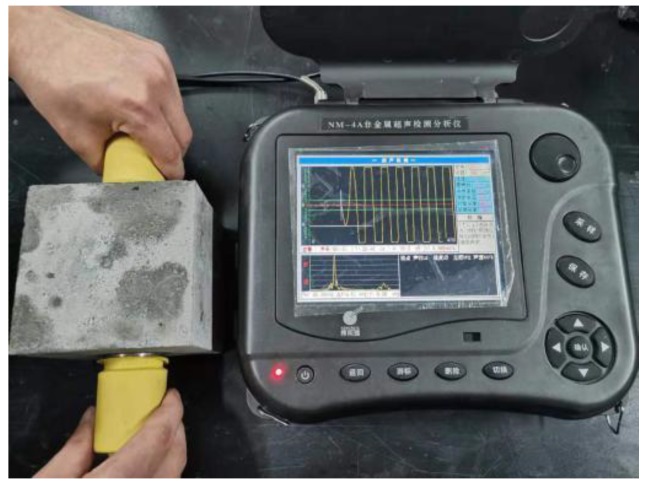
Ultrasonic test diagram.

**Figure 2 materials-13-01830-f002:**
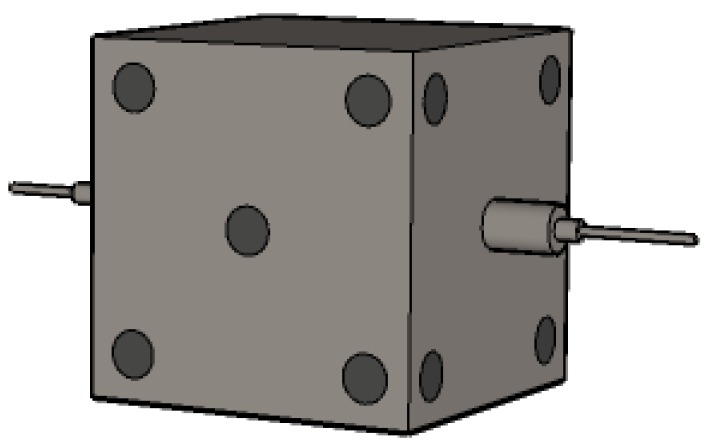
Simulation diagram of ultrasonic test.

**Figure 3 materials-13-01830-f003:**
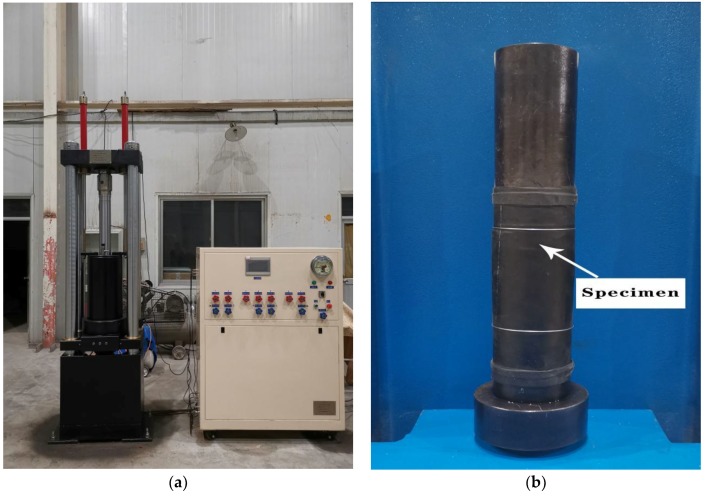
Loading device and specimen: (**a**) Loading device; (**b**) Specimen wrapped with thermoplastic pipe.

**Figure 4 materials-13-01830-f004:**
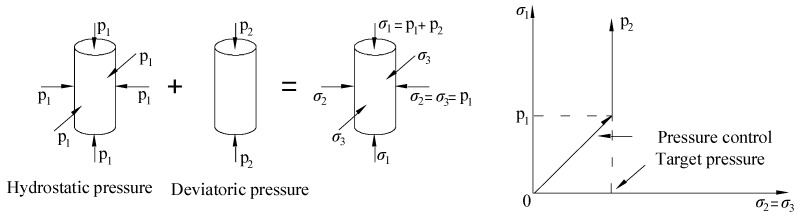
Stress model and loading path of specimen.

**Figure 5 materials-13-01830-f005:**
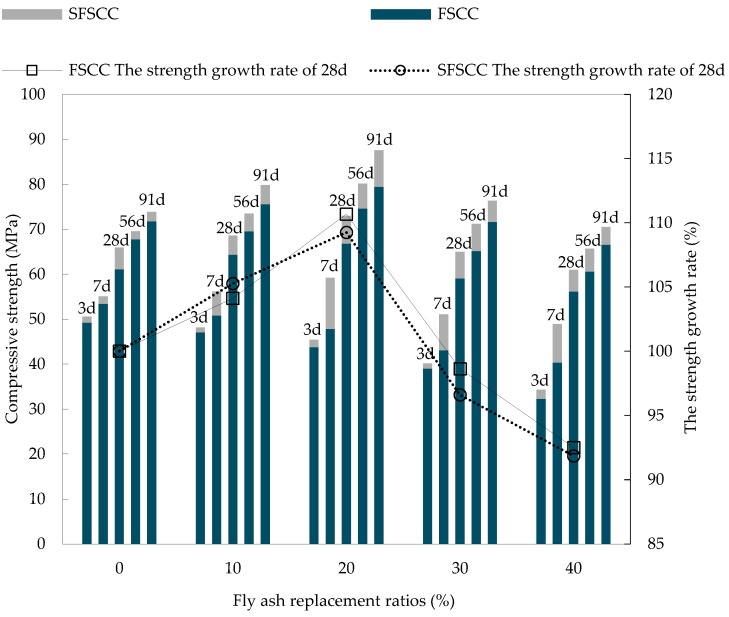
Results of SCC compressive strength test.

**Figure 6 materials-13-01830-f006:**
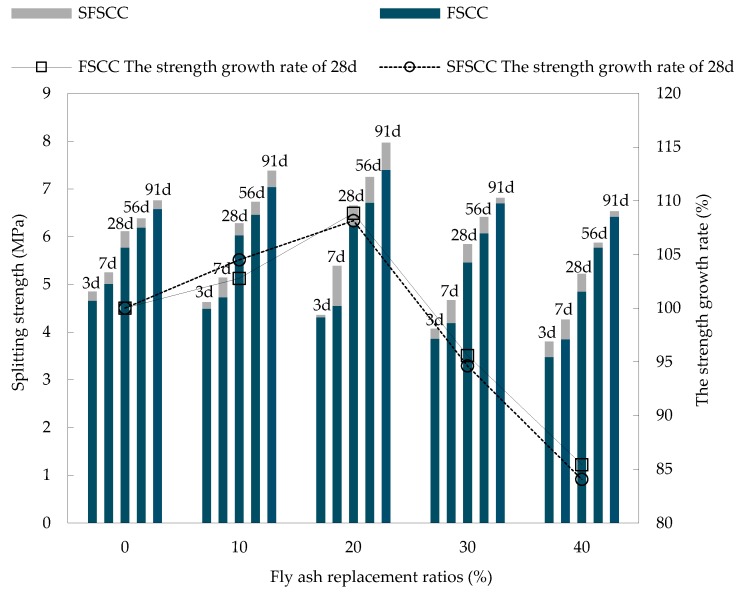
Results of SCC splitting strength test.

**Figure 7 materials-13-01830-f007:**
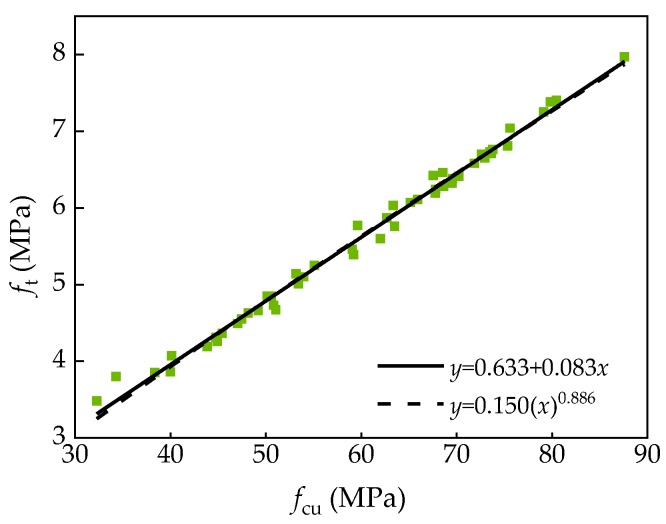
Relationship between compressive strength and splitting strength.

**Figure 8 materials-13-01830-f008:**
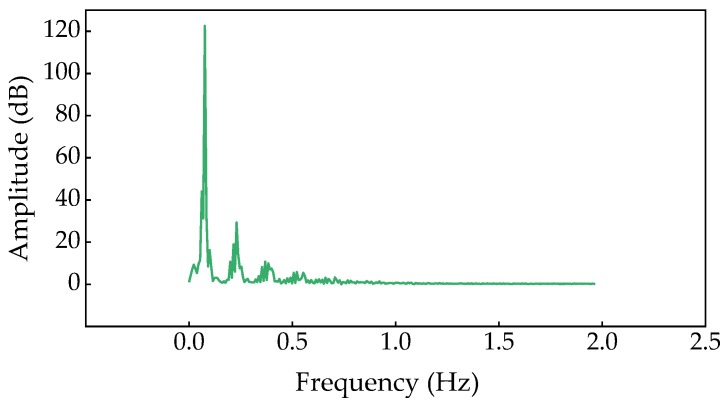
Spectrum of SFSCC20.

**Figure 9 materials-13-01830-f009:**
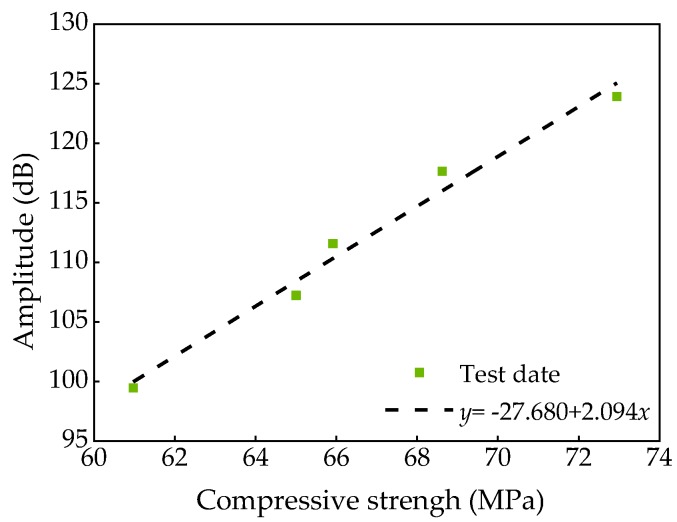
Relationship between amplitude and compressive strength.

**Figure 10 materials-13-01830-f010:**
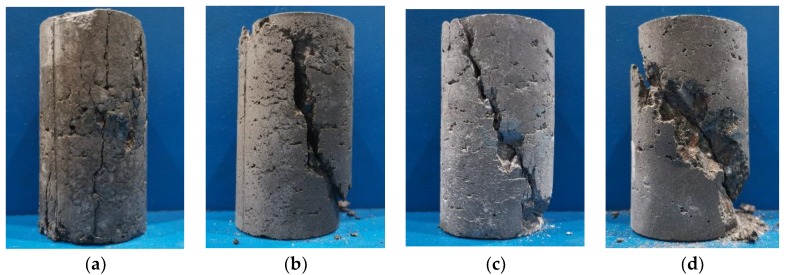
Failure mode of specimens under different confining pressures: (**a**) *σ*_2_ = *σ*_3_ = 0 MPa; (**b**) *σ*_2_ = *σ*_3_ = 5 MPa; (**c**) *σ*_2_ = *σ*_3_ = 10 MPa; (**d**) *σ*_2_ = *σ*_3_ = 15 MPa.

**Figure 11 materials-13-01830-f011:**
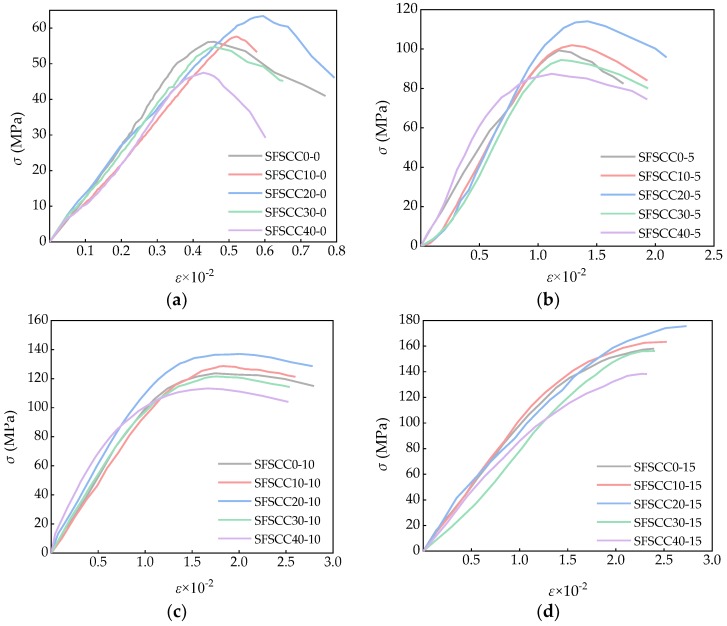
Stress-strain curve of part specimens: (**a**) *σ*_2_ = *σ*_3_ = 0 MPa; (**b**) *σ*_2_ = *σ*_3_ = 5 MPa; (**c**) *σ*_2_ = *σ*_3_ = 10 MPa; (**d**) *σ*_2_ = *σ*_3_ = 15 MPa.

**Figure 12 materials-13-01830-f012:**
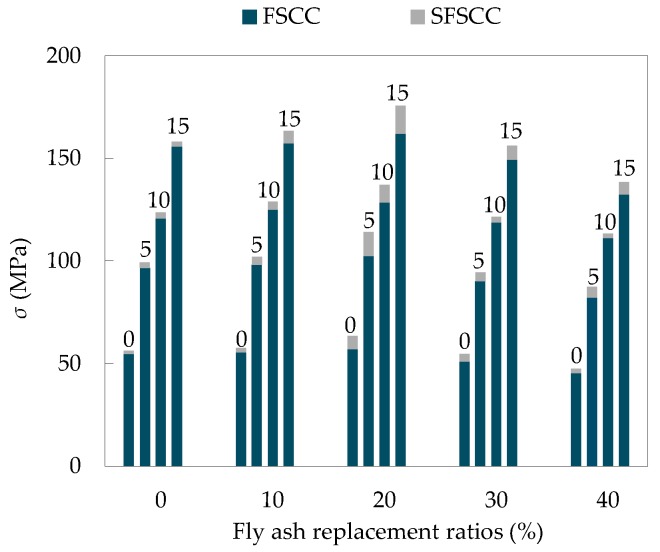
SCC peak stress under different confining pressures.

**Figure 13 materials-13-01830-f013:**
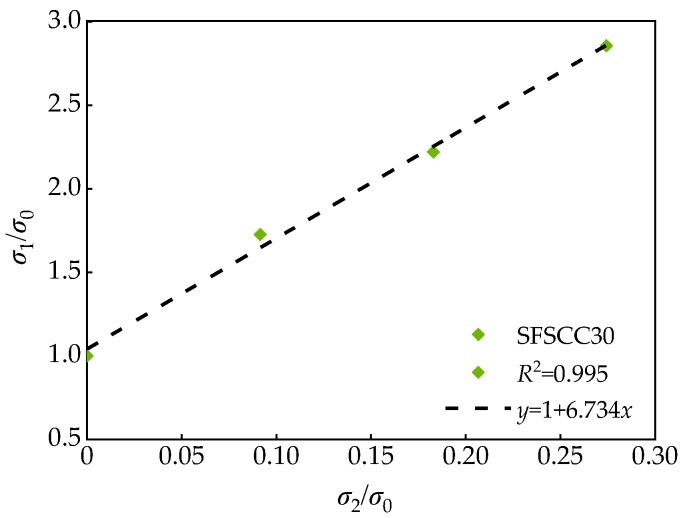
Relationship between peak stress and confining pressure of SFSCC30.

**Figure 14 materials-13-01830-f014:**
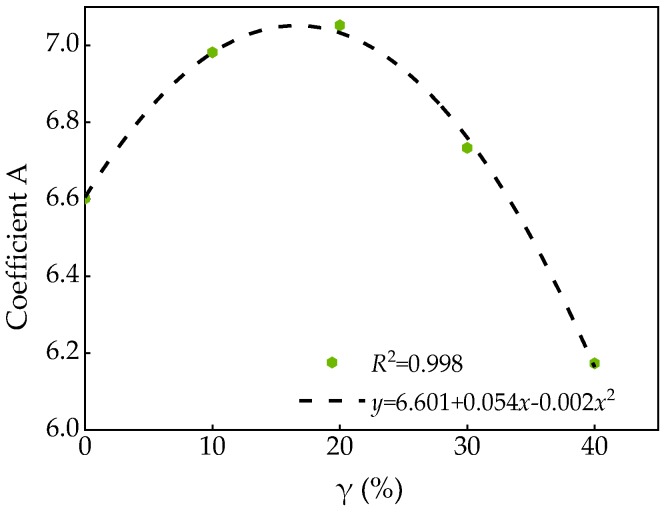
Relationship between coefficient A and FA substitution rate γ.

**Figure 15 materials-13-01830-f015:**
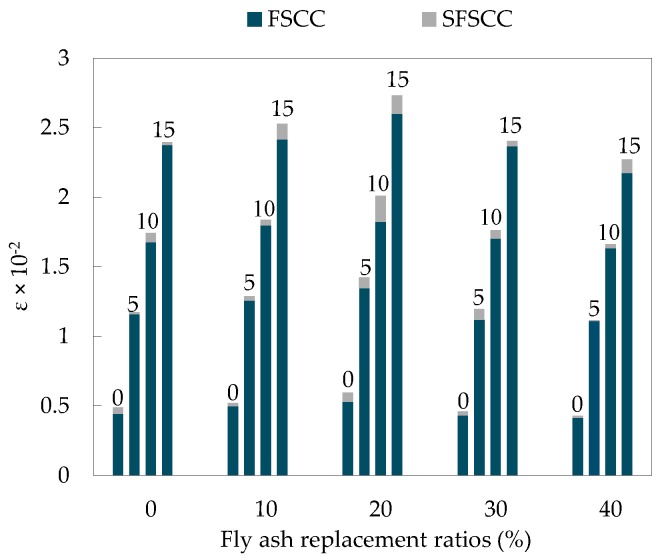
Peak strain of SCC under different confining pressures.

**Figure 16 materials-13-01830-f016:**
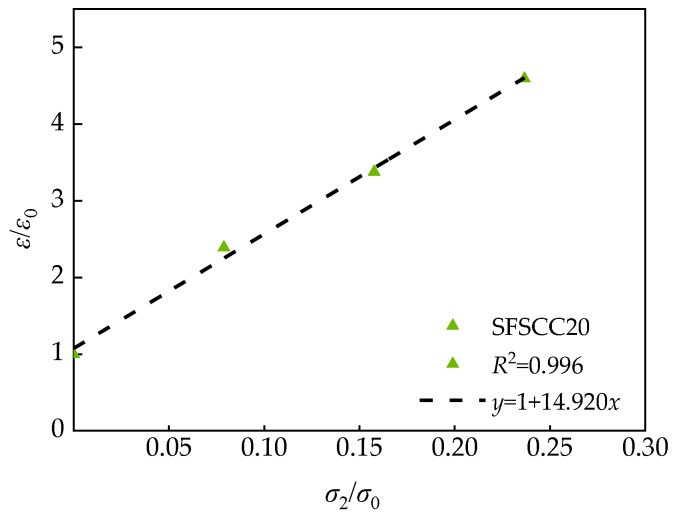
Relationship between peak strain and confining pressure of SFSCC20.

**Figure 17 materials-13-01830-f017:**
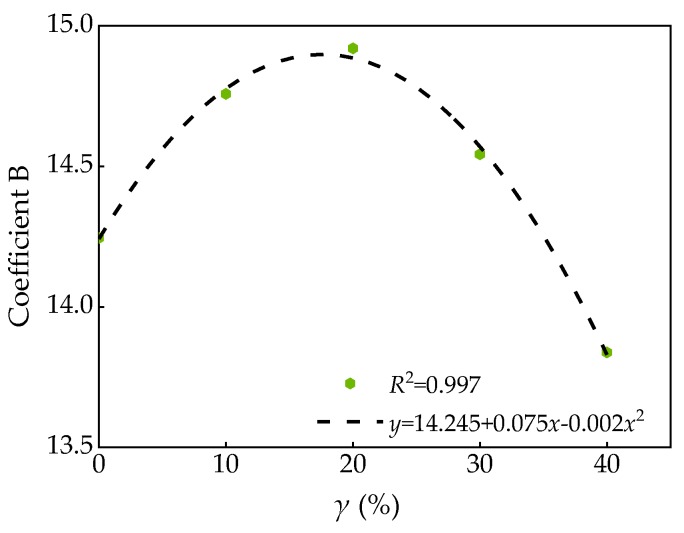
Relationship between coefficient B and FA substitution rate *γ*.

**Figure 18 materials-13-01830-f018:**
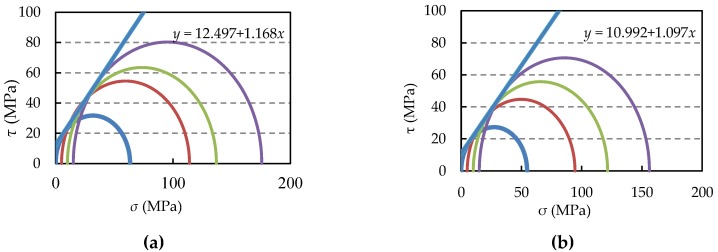
Envelope of SCC partial stress circle: (**a**) SFSCC20; (**b**) SFSCC30.

**Figure 19 materials-13-01830-f019:**
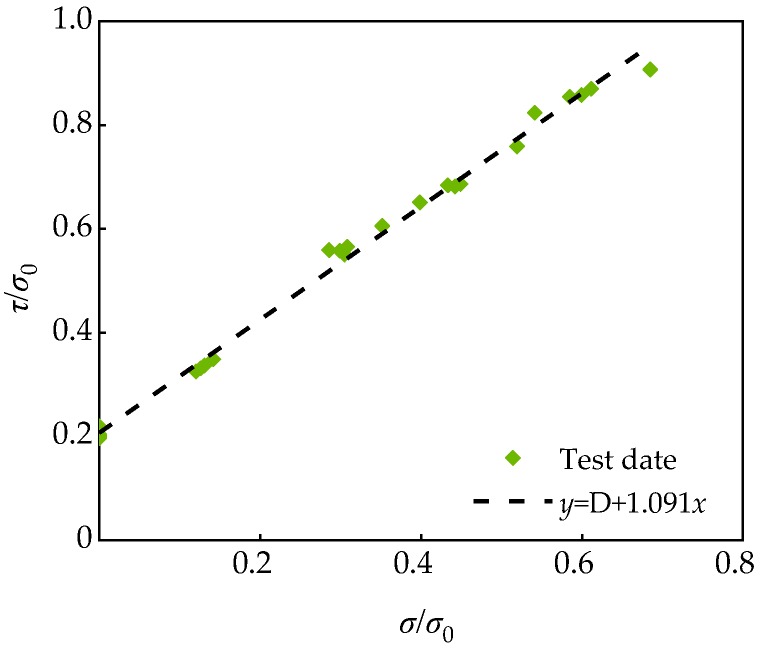
Relationship between *τ*/*σ*_0_ and *σ*/*σ*_0_.

**Figure 20 materials-13-01830-f020:**
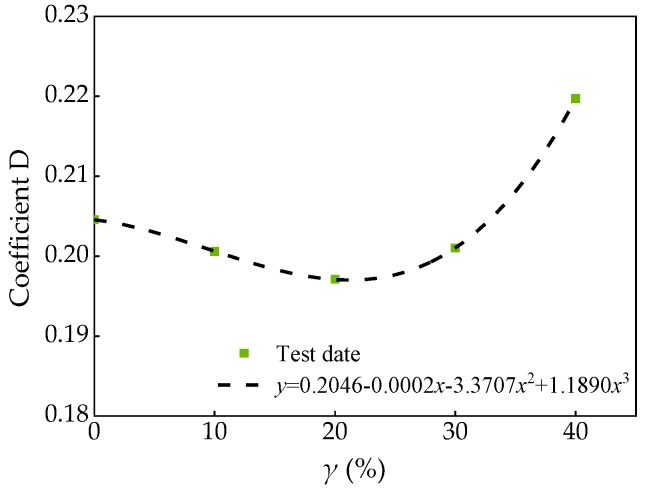
Relationship between coefficient D and substitution rate *γ*.

**Table 1 materials-13-01830-t001:** The main performance indexes of cement and FA.

Chemical Composition (%)	Cement	FA	Physical Index	Cement	FA
SiO_2_	22.46	40.90	Fineness (%)	1.60	17.9
SO_3_	2.96	2.96	Loss on ignition (%)	1.40	2.08
CaO	57.15	9.40	Water demanded (%)	−	97
K_2_O	0.86	2.80	Setting time (min)	−	−
MgO	1.54	4.16	Initial setting	200	−
Al_2_O_3_	7.60	23.70	Final setting	245	−
Na_2_O	0.31	−	Compressive strength (MPa)	−	−
Fe_2_O_3_	5.00	11.30	3 days	25.48	−
TiO_2_	−	0.16	28 days	42.67	−

**Table 2 materials-13-01830-t002:** The main performance indexes of SF.

Chemical Composition (%)							Density (kg/m^3^)	28 days Activity Index (%)
SiO_2_	Al_2_O_3_	Fe_2_O_3_	K_2_O	CaO	MgO	Na_2_O		
92.00	0.30	0.80	0.90	0.40	0.30	0.20	18,000	95

**Table 3 materials-13-01830-t003:** Mix Proportion of SCC (kg/m^3^).

Specimen Number	Water	Cement	FA	SF	Gravel	Sand	Expansive Agent	Water Reducer
FSCC0	162.43	597.16	0	0	867.20	790.19	47.77	7.16
FSCC10	162.43	537.44	59.72	0	867.20	790.19	47.77	7.16
FSCC20	162.43	477.73	119.43	0	867.20	790.19	47.77	7.16
FSCC30	162.43	418.01	179.15	0	867.20	790.19	47.77	7.16
FSCC40	162.43	358.29	238.87	0	867.20	790.19	47.77	7.16
SFSCC0	162.43	573.27	0	23.89	867.20	790.19	47.77	7.16
SFSCC10	162.43	513.55	59.72	23.89	867.20	790.19	47.77	7.16
SFSCC20	162.43	453.84	119.43	23.89	867.20	790.19	47.77	7.16
SFSCC30	162.43	394.12	179.15	23.89	867.20	790.19	47.77	7.16
SFSCC40	162.43	334.40	238.87	23.89	867.20	790.19	47.77	7.16

**Table 4 materials-13-01830-t004:** Test results of 3 days, 7 days and 28 days compressive strength of SCC.

Specimen Number	The 3 days strength (MPa)			The 7 days strength (MPa)			The 28 days strength (MPa)		
	Test Results	Average	STD	Test Results	Average	STD	Test Results	Average	STD
FSCC0	45.76/50.18/51.66	49.2	2.50	56.81/44.93/53.42	51.72	4.50	66.06/57.36/62.97	62.13	3.60
FSCC10	42.81/52.21/46.10	47.04	3.89	54.26/46.06/52.11	50.81	3.47	62.96/69.99/60.10	64.35	4.16
FSCC20	43.78/36.77/46.41	42.32	4.07	44.73/50.43/48.42	47.86	2.36	69.71/61.67/68.99	66.79	3.63
FSCC30	41.82/35.23/38.92	38.99	2.71	47.74/38.89/42.49	43.04	3.63	59.05/67.93/55.51	60.83	5.22
FSCC40	34.31/30.71/31.79	32.27	1.51	42.07/37.31/41.67	40.35	2.16	65.31/56.15/54.13	58.53	4.86
SFSCC0	51.28/54.05/46.32	50.55	3.20	50.58/59.45/55.27	55.10	3.62	68.40/59.86/69.47	65.91	4.30
SFSCC10	52.77/46.71/44.97	48.15	3.34	47.96/56.17/59.32	54.48	4.79	65.03/72.68/68.15	68.62	3.14
SFSCC20	43.75/48.92/43.62	45.43	2.47	57.41/61.39/58.80	59.20	1.65	68.33/79.24/71.28	72.95	3.41
SFSCC30	46.69/40.13/39.67	42.16	3.21	42.47/54.50/51.04	49.27	5.06	73.97/65.00/64.56	67.98	4.34
SFSCC40	35.34/36.76/30.74/	34.28	2.57	52.29/49.03/45.35	48.89	2.83	60.62/65.18/57.11	60.97	3.30

**Table 5 materials-13-01830-t005:** Test results of 56 days and 91 days compressive strength of SCC.

Specimen Number	The 56 days Strength (MPa)			The 91 days Strength (MPa)		
	Test Results	Average	STD	Test Results	Average	STD
FSCC0	66.91/62.76/73.61	67.76	4.47	68.42/75.55/71.37	71.78	2.93
FSCC10	64.54/71.51/72.60	69.55	3.57	68.55/78.85/79.37	75.59	4.98
FSCC20	68.43/77.34/78.09	74.62	4.39	82.46/72.59/83.3	79.45	4.86
FSCC30	67.66/62.86/64.90	65.14	1.97	76.57/67.08/71.24	71.63	3.88
FSCC40	70.51/60.61/60.38	63.83	4.72	66.13/71.18/62.31	66.54	3.63
SFSCC0	74.83/64.79/69.09	69.57	4.11	74.85/63.29/74.49	70.88	5.37
SFSCC10	73.51/62.21/74.33	70.02	5.53	76.19/80.55/82.69/	79.81	2.70
SFSCC20	84.41/74.65/81.33	80.13	4.07	81.19/90.73/90.83	87.59	4.52
SFSCC30	68.86/70.29/74.51	71.22	2.40	82.42/71.49/75.08	76.33	4.55
SFSCC40	70.21/62.80/64.03	65.68	3.24	72.96/67.86/70.74	70.52	2.09

**Table 6 materials-13-01830-t006:** Test results of 3 days, 7 days and 28 days splitting strength of SCC.

Specimen Number	The 3 days Strength (MPa)			The 7 days Strength (MPa)			The 28 days Strength (MPa)		
	Test Results	Average	STD	Test Results	Average	STD	Test Results	Average	STD
FSCC0	4.83/4.51/4.64	4.66	0.13	4.56/5.18/5.29	5.01	0.32	5.89/5.26/6.16	5.77	0.38
FSCC10	4.11/4.65/4.71	4.49	0.27	4.62/4.98/4.59	4.73	0.18	6.24/5.60/6.25	6.03	0.30
FSCC20	4.15/4.54/4.24	4.31	0.17	4.49/4.86/4.30	4.55	0.23	6.02/6.78/5.92	6.24	0.38
FSCC30	3.86/4.48/4.01	4.12	0.26	4.55/3.95/4.07	4.19	0.26	5.46/5.95/4.97	5.46	0.40
FSCC40	3.41/3.75/3.28	3.48	0.20	3.57/4.13/3.85	3.85	0.23	5.21/4.73/4.61	4.85	0.26
SFSCC0	5.03/4.53/4.99	4.85	0.23	5.55/5.02/5.18	5.25	0.22	6.08/5.90/6.35	6.11	0.18
SFSCC10	4.97/4.28/4.64	4.63	0.28	5.96/5.14/4.83	5.31	0.48	6.11/6.57/6.16	6.28	0.21
SFSCC20	4.50/4.42/4.16	4.36	0.15	5.60/5.03/5.54	5.39	0.26	6.79/6.35/6.81	6.65	0.21
SFSCC30	4.15/4.24/3.82	4.07	0.18	4.95/4.38/4.68	4.67	0.23	5.93/6.01/5.58/	5.84	0.19
SFSCC40	4.46/3.65/3.8	3.97	0.35	4.27/4.46/4.04/	4.26	0.17	4.32/5.22/5.34	4.96	0.46

**Table 7 materials-13-01830-t007:** Test results of 56 days and 91 days splitting strength of SCC.

Specimen Number	The 56 days Strength (MPa)			The 91 days Strength (MPa)		
	Test Results	Average	STD	Test Results	Average	STD
FSCC0	6.51/5.69/6.37	6.19	0.36	6.76/6.15/6.83	6.58	0.31
FSCC10	6.89/6.04/6.45	6.46	0.35	7.27/7.13/6.72	7.04	0.23
FSCC20	6.32/6.86/6.95	6.71	0.28	7.58/6.89/7.73	7.40	0.51
FSCC30	6.14/5.12/6.07	5.78	0.47	6.63/6.46/7.01	6.70	0.23
FSCC40	6.18/5.40/5.73	5.77	0.32	5.87/6.66/6.73	6.42	0.39
SFSCC0	6.27/6.45/6.42	6.38	0.08	6.32/7.79/6.76	6.96	0.62
SFSCC10	6.46/6.9/6.83	6.73	0.19	7.61/6.91/7.62	7.38	0.33
SFSCC20	6.74/7.40/7.61	7.25	0.37	8.15/7.45/8.31	7.97	0.37
SFSCC30	6.22/6.82/6.19	6.41	0.29	6.98/6.44/7.01	6.81	0.26
SFSCC40	5.93/5.52/6.16	5.87	0.26	6.71/6.07/6.81	6.53	0.33

**Table 8 materials-13-01830-t008:** Fitting relationship of SCC.

Specimen Number	*γ* (%)	Coefficient A	Fitting Relation	*R* ^2^
SFSCC0	0	6.601	$\frac{\sigma_{1}}{\sigma_{0}}=1+6.601\frac{\sigma_{2}}{\sigma_{0}}$	0.989
SFSCC10	10	6.982	$\frac{\sigma_{1}}{\sigma_{0}}=1+6.982\frac{\sigma_{2}}{\sigma_{0}}$	0.990
SFSCC20	20	7.052	$\frac{\sigma_{1}}{\sigma_{0}}=1+7.052\frac{\sigma_{2}}{\sigma_{0}}$	0.980
SFSCC30	30	6.734	$\frac{\sigma_{1}}{\sigma_{0}}=1+6.734\frac{\sigma_{2}}{\sigma_{0}}$	0.995
SFSCC40	40	6.173	$\frac{\sigma_{1}}{\sigma_{0}}=1+6.173\frac{\sigma_{2}}{\sigma_{0}}$	0.986

**Table 9 materials-13-01830-t009:** Fitting relationship of SCC.

Specimen Number	*γ* (%)	Coefficient B	Fitting Relation	*R* ^2^
SFSCC0	0	14.245	$\frac{\sigma_{1}}{\sigma_{0}}=1+14.245\frac{\sigma_{2}}{\sigma_{0}}$	0.999
SFSCC10	10	14.758	$\frac{\sigma_{1}}{\sigma_{0}}=1+14.758\frac{\sigma_{2}}{\sigma_{0}}$	0.996
SFSCC20	20	14.920	$\frac{\sigma_{1}}{\sigma_{0}}=1+14.920\frac{\sigma_{2}}{\sigma_{0}}$	0.996
SFSCC30	30	14.543	$\frac{\sigma_{1}}{\sigma_{0}}=1+14.543\frac{\sigma_{2}}{\sigma_{0}}$	0.998
SFSCC40	40	13.838	$\frac{\sigma_{1}}{\sigma_{0}}=1+14.838\frac{\sigma_{2}}{\sigma_{0}}$	0.998

**Table 10 materials-13-01830-t010:** Mohr-Coulomb parameters of SCC with FA with SF.

Specimen Number	*C* (MPa)	*φ* (°)
SFSCC0	11.488	47.653
SFSCC10	11.558	48.421
SFSCC20	12.497	49.430
SFSCC30	10.992	47.644
SFSCC40	10.429	45.738

**Table 11 materials-13-01830-t011:** Relation of SCC stress circle envelope function.

Specimen Number	*γ* (%)	Function Relation
SFSCC0	0	y=11.488+1.097x
SFSCC10	10	y=11.558+1.127x
SFSCC20	20	y=12.497+1.168x
SFSCC30	30	y=10.992+1.097x
SFSCC40	40	y=10.429+1.026x

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
