# Peer review of "Experimental Investigation on the Mechanical Properties of Self-Compacting Concrete under Uniaxial and Triaxial Stress"

_materials, 2020, doi:10.3390/ma13081830_

Round 1

Reviewer 1 Report

The paper discusses the effect of FA and SF addition to SCC mixtures.

The novelty of the paper must be stressed. The effects reported is quite expected (even authors themselves several times use the term: obvious), and the added value of the should be better highlighted in the introduction section.

  • GFSCC is SFSCC throughout the manuscript
  • Line 212: it is from literature (please cite the source) or it is an evidence from your experiments? Please clarify this point
  • Lines 221-222, and line 297: (the same) are there equations proposed by the authors? In this case, what is their physical meaning? It is only a pure mathematical model or it is based on any physical-chemical consideration?
  • In this kind of papers, a robust statistical analysis is required: how many replicate tests were carried out?

Author Response

Dear Reviewer:

Thanks very much for your hard work. We have revised the manuscript according to your kind advice and referee’s detailed advice, and the amendments are highlighted in red in the revised manuscript. We sincerely hope this manuscript will be finally acceptable to be published. Thanks very much for all your help and looking forward to hearing from you soon.

Best regards

Sincerely yours

Reviewer 2 Report

This study attempts to investigate impact of adding fly ash and silica fume into mechanical properties of self-compacting concrete. Authors have examined variation of compression, tension and triaxial behaviour. The work presented here is novel enough to publish and will provide a worthwhile contribution to the literature. However, further explanations are required in some areas before it published. The conclusion section can be expanded while stick to the key research findings.

Specific comments are:

Need to expand introduction section while adding upto date published reviews based on self-compacting concrete behaviour. I can see some important publications are missing there and required to clearly report the current gap of previous works, and how you going to address them in your study. Clearly define the research scope.

Report mineralogical properties and particle size distribution of fly ash which plays significant role reaction. Can properties of concrete change have based on fly ash location/power station, if so upto what extent?

Authors need to mark standard deviation for all test results

Suggest comparing mechanical relationships with relevant standards such as ACI codes, FIB codes etc.

Why compression/tensile strengths will reduce when exceeding fly ash replacement in self-compacting concretes?

Author Response

(The authors gave the same response as above.)
